# The Impact of Institutional Quality on Sectoral Foreign Direct Investment in Pakistan: A Dynamic Simulated ARDL Approach

Zafar Ullah Khan [1], Alam Khan [1,*], Dilawar Khan [1,*] and Róbert Magda [2,3]

[1] Department of Economics, Kohat University of Science & Technology, Kohat 26000, Pakistan; tozafar1@yahoo.com
[2] Hungarian National Bank—Research Center, John von Neumann University, 6000 Kecskemét, Hungary; rmagda72@gmail.com
[3] Vanderbijlpark Campus, Northwest University, Vanderbijlpark 1900, South Africa
[*] Correspondence: alamkhan@kust.edu.pk (A.K.); dilawar@kust.edu.pk (D.K.)

**Abstract:** Capital is needed to accelerate the development process, but developing nations struggle with minimum resources. Since foreign direct investment (FDI) is essential to solve the funding shortage, developing nations make every effort to attract FDI to their countries. This study was conducted to explore the impact of institutional quality on sectoral FDI in Pakistan using time series data for the period from 1986 to 2019. To create a solid foundation for policy formulation, this study developed a single measure of institutional quality by utilizing a wide range of institutional indicators. It evaluates the impact of overall institution quality on sector-level FDI and examines the causal relationship between institutions and sectoral FDI with a clear focus on a single-country analysis. The dynamic simulated autoregressive distributed lag technique was employed to explore the impact of institutional quality and other factors on FDI. The results of the study explained that institutional quality and TO have a positive and significant effect on the FDI of the primary FDI sector at 5%, while in the case of the secondary sector, the effect of institutional quality, HDI and GDP on FDI inflow is significant at 10% and TO has a significant effect on secondary sector FDI at 5%. In addition, institutional quality and GDP have a positive relationship with tertiary sector FDI at 5%. The empirical findings show that higher institutional quality in emerging economies such as Pakistan encourages large transfers of technological advances through FDI, increasing the overall performance of the economy. This study found that institutional quality significantly increases sectoral FDI in Pakistan. Finally, this study prescribes some policy measures to increase FDI based on the findings.

**Keywords:** institutional quality; foreign direct investment; sectoral FDI; dynamic simulated ARDL model; Pakistan

## 1. Introduction

Foreign direct investment is considered an important variable in economic growth in developing countries. The most significant problem in almost all emerging nations is the lack of capital, which has a direct impact on investment because it is needed along with other elements of production. Over time, human priorities change, requiring the introduction of new technologies made possible by FDI. Foreign direct investments are normally made through multinational corporations. With this, the local market becomes more competitive and national companies can produce at their maximum capacity, which reduces costs and favors exports. With the exception of a few, most fast-growing nations did so through FDI. Foreign direct investment plays an important role in the development of developing nations, which has been highlighted in recent research [1]. The most significant issue almost all emerging nations have is a lack of capital, which has a direct impact on investment because it is necessary along with other production elements. As time goes on, human priorities shift, necessitating the introduction of new technology, which is made feasible by FDI. Foreign direct investments are typically made through multinational

corporations. As a result, the local market becomes more competitive and domestic firms can produce at their maximum capacity, which reduces costs and promotes exports [2]. Because FDI helps developing nations grow faster and reduces unemployment, they are keenly interested in luring that kind of investment to their home markets. It is challenging for countries to achieve desired growth rates and a decline in the level of poverty when there is an insufficient influx of investment. To provide a complete view of the total FDI influx to Pakistan, FDI is examined in several different industries. Prior to the 1990s, mining and quarrying had a sizable share of the economy compared to other sectors. When FDI flow is examined in the economy, the clear position in each sector is not explained, but when it is examined in isolation, the weak areas are identified. The decision-makers then concentrate on the weaker segments of the country and work to direct more foreign money to those areas where it is most needed. The primary sector includes a variety of industries, including those that use tobacco, food, drinks, rubber as a raw material and products derived from rubber, paper, pulp, sugar, and leather remnants and finished goods. Industrial groups from the secondary sector include those that produce cement, petrochemicals, machinery, various metals, electronics, metal products, power, electrical equipment, transport equipment, oil and gas exploration, mining, and quarrying. Similarly, the fore tertiary sector includes tourism, wholesale and retail trade, financial services, storage, transportation, and communication, as well as private and social services [3].

Developing nations such as Pakistan face a deficiency of capital in the domestic market to fulfill investment requirements. These countries are working to solve some fundamental issues to promote human development and lower poverty rates. There are various prerequisites that must be met to achieve these goals, with capital being the most important one. Investment in capital raises per capita income for the populace and lowers the economy's poverty rate. People's living standards rise when their own discretionary income rises, making it possible for them to acquire high-quality health care and education. In most cases, FDI inflows are taken for granted as a kind of capital investment. As FDI inflows have surged in recent decades, countries' HDI and per capita GDP have also risen [4]. The researchers discovered a significant link between FDI and economic growth. Countries strive to create an atmosphere that is beneficial from an economic standpoint to draw in as much FDI as possible since it speeds up the development process [5]. Institutional quality is viewed from a business perspective as a highly important feature that draws FDI into the economy, which is crucial for the growth of the local market. The development process benefits from high-quality institutions, especially in regard to luring in more foreign investment [6]. Due to massive development initiatives, developing nations have recently formed a race to draw FDI. Institutions have an impact on economic growth in emerging nations: they cause a variety of issues, such as a lack of investment, a decline in productivity, a reduction in population income, and a decline in national output. On the other hand, well-organized institutions effectively allocate resources across various industries and direct investment toward high-yield ventures [7]. Human development entails expanding a person's options so that they can enjoy a longer life with excellent health, obtain access to high-quality education, and raise their level of living [8]. In addition, it covers political freedom, self-respect, and human rights.

Considering the foregoing, this study created a specific measure of institutional quality by using a broad range of institutional indicators that evaluate the impact of generally high-quality institutions on sector-level FDI and observe the causal relationship between institutions and sector-level FDI with the utmost focus on a single-country examination to produce impressive findings for further consideration in policy recommendation. In our review of the literature, we found that no reliable and trustworthy studies had been published that explored the possibility of a causal relationship between institutional quality and sector-level FDI, including FDI in primary services and manufacturing. The goal of the current study is to examine the long- and short-run causal relationships between institutional quality and sector-specific FDI in Pakistan using the dynamic simulated autoregressive distributive leg (DSARDL) technique. This study examines what makes

institutional quality FDI more sustainable? The most recent ARDL model, which is infrequently used in the study in question, is this one. The advantage of this model was that it could resolve the estimation problems with the definition of the simple ARDL model in the short run and long run. This suggested paradigm has the benefit of estimating, stimulating, and robotically anticipating counterfactual changes in one explanatory variable and their consequences on explained variables while keeping other control variables constant [9,10]. The model generates, draws, and displays graphs of the variables' predicted positive and negative variance as well as their short- and long-term associations automatically. Only the relationship between short-run and long-run variables may be estimated using the ARDL model created by [11].

Pakistan is a heavily populated nation. According to statistics, its population growth rate is very high. Pakistan places fifth in the list of most populated countries all over the world, according to the results of the 2017 census. It requires generating employment at the same rate at which its population is growing. A single country cannot produce all of its required commodities, so it enters international trade, but international transactions take place in the medium of foreign reserves. To increase employment in the country, investment needs capital in the domestic market, and developing countries such as Pakistan face capital deficiency. Therefore, FDI inflow is compulsory to cover the space of domestic investment deficiency. Foreign investment creates job opportunities by expanding business activities at the domestic level and increasing exports from the host country. The increase in exports is a source of capital formation for the country. The literature discloses that INQ has a significant role in affecting FDI inflow in a country. The structure of INQ creates certainty in the business environment, while uncertainty directly discourages foreign investment inflow in the host country. MNCs take interest in developing countries that introduce INQ reforms, which means that the poor structure of institutions disappoints foreign investors. This research covers the important variables under the INQ that play an important role in drawing FDI in Pakistan.

Pakistan receives FDI inflow mainly in two dimensions: one is the government preference to encourage foreign investment in specific areas and the other is that MNCs' priority mostly depend on the profit factor. Pakistan is a large country, considered number seven in the world. Pakistan has abundant natural resources and favorable geographic location and weather, and other factors to attract FDI inflow, but could not increase the foreign capital inflow in the economy, while its neighboring countries increased foreign capital inflow to an effective level. At the start of the 1990s, Pakistan introduced policy reforms that enhance the inflow of capital to the country, but it did not continue in the following years because of political instability. In the start of first decade of this century, international peace was destroyed due to the 9/11 attack and war against terrorism was started. Pakistan served as a frontline state against the terrorism and faced a huge loss in the form of attracting FDI in the international market. Pakistan's share was 0.06% of total international FDI till 2012, and the US, UAE and UK are the conventional investors in Pakistan. In the 1980s era, mostly MNCs were interested to invest in mining and manufacturing industries, both sectors sharing 81% of total FDI inflow, but FDI inflow in each sector decreased to 30% during 1995–1999 and increased to 49% in 2000–2004. In later years, the FDI shifted to transport and communication, human services and storage sectors in Pakistan, which shows FDI inflows shifting from one sector to another with specific preferences in the previous years. After the reform period, FDI changed sector-wise, with the share of the services sector increasing by 2.2% in 1980–1995 and touching 45% in 1995–2010. Pakistan is facing poor institutional quality when compared to other LDCs in near region [12].

This study examines how institutional quality makes FDI more sustainable. Pakistan is working hard to increase its share of world FDI. INQ affects FDI in different regions of the world. This study examines the impact of INQ on the inflow of foreign investment in Pakistan, analyzing different dimensions included in INQ. This research explains the effects of INQ on sector-wise FDI. It will help the policy-makers to provide exact knowledge of all sectors and make economic plans to overcome the problem of FDI inflow in the country.

Furthermore, this study will help foreign investors in showing the status of FDI inflow in all sectors in Pakistan. In this way, resources will move to the sector that receives less foreign capital and the country will achieve self-sufficiency.

Furthermore, this study developed a particular measure of institutional quality by applying a wide set of institutional indicators that assess the impact of the overall quality of institutions at sector-level FDI and observe the causal effect between institutions and sectoral FDI with a perfect emphasis on a single-country investigation to achieve significant results for further use in policy recommendation. In the literature review, we observed that no sound, authentic work has been published that examined causality between institutional quality and sector-level FDI, including primary services and manufacturing FDI. The present study employs the dynamic simulated autoregressive distributive leg (DSARDL) approach to investigate the long- and short-run causal relationships between institutional qualities and sectoral FDI in Pakistan. This study examines the impact of institutional quality on sector-wise FDI in Pakistan, which includes three sectors (primary, secondary, and tertiary). Although FDI is crucial for creating new job possibilities in the economy, which is desirable from a government perspective for national economies, several studies have revealed that FDI has both positive and negative effects on the labor market. Finding the various factors that are positively impacted by FDI should come before looking at the impact on employment levels. The demand for labor rises when this kind of capital enters the domestic market, and if it is invested in labor-intensive industries, it will play a significant role. Additionally, it will boost the capability of local industry employment that results from forward or backward links.

The remainder of the article is structured as follows. The literature review is discussed in the second section of the paper. The third and fourth sections provide the research methodology and a discussion of the findings, respectively. The conclusion and policy recommendations are covered in the final part of the manuscript.

## 2. Literature Review

The empirical findings show a positive correlation between the ability of the government to manage corruption, the standard of its legislation, the degree of voice and accountability, and the attraction of FDI to African nations [13]. Another study investigated the impact of institutions on foreign direct investment locale preferences in India using a novel data set on two-sided FDI [14]. That study supports other studies by asserting that institutions and FDI have a beneficial relationship. The public judicial system, the bureaucracy structure, and asset-ownership rights security are three widely used institutional core factors that are used to study the influence of institutions on FDI placement decisions. The gravity model is used to build on earlier research that focused on the impact of firm heterogeneity on the connection between institutions and FDI. Institutions, as per the report, have a significant and advantageous influence on FDI destination choice. Institutions in developed regions have a good relationship with FDI. Unexpectedly, a negative correlation between FDI and institutions in emerging regions was found. The enterprises were divided into four groups according to size to capture their heterogeneity. Large firms are more likely to invest, according to the findings, in nations with solid institutions. Reference [15] uses the feasible generalized least squares (FGLS) method to examine the effects of FDI inflows, institutional quality (IQ), and their relationships on income disparities in 36 Asian countries between 2000 and 2018. The results demonstrate that FDI exacerbates economic disparity and that a rise in IQ from FDI lessens this adverse effect up to a certain IQ level, at which point FDI reduces income variation. Additionally, reduced income inequality is a result of enhanced institutional quality, and the positive impact of increasing FDI inflows is amplified. In particular, the effectiveness of public administration, the reduction of corruption, the stability of the government, the absence of terrorism, and law enforcement regulate the impact of FDI on income inequality. Study [16] was conducted to quantify the impact of some macroeconomic factors on FDI inflows. For the ASEAN-7, panel data analysis showed that tax burden, inflation, economic institution quality, and economic growth are significant

elements that drag FDI, although political institutions and population growth reduce FDI. The study suggested result-based policies to lift FDI inflow. Study [17] highlighted the impact of institutional quality and corruption on the shadow economy. The results based on 29 Asian countries reveal that corruption increases, institution quality decreases and the interaction between economic freedom and corruption negatively affects the shadow economy. Furthermore, democracy has a positive association with the shadow economy. Similarly, another study was conducted to examine the effectiveness of the audit committee on the stability of banks. The results interestingly disclose the fact that a smaller committee with independent members improves bank stability. It is important that the relationship between the effectiveness of the committee and bank stability be contingent on the financial condition of the bank and institutional quality [18]. Moreover, Ref. [19] wanted to inspect the presence of relationships at various stages between bank risk and risk governance effectiveness. The analysis explored a negative association in the developed world with better institutional quality.

Pakistan also depends on FDI inflow to overcome the shortage of capital in the domestic market. FDI inflow increased during 2007–2008, but declined in later years. The allure of FDI appears to have faded since international investment and portfolio investments have steadily dropped in recent years. This decrease in FDI intake necessitates an investigation of the variables that are causing this discouraging trend in FDI influx. The inflow of FDI serves a key function in accelerating the economy's engine to operate government affairs smoothly and effectively. The quality of the host country's institutions, macroeconomic indices, and natural resources all influence the influx of FDI into developing countries. The importance of organizations and institutional quality in affecting FDI inflows cannot be overstated. Economic performance is generally observed to be better in countries with upgraded institutions, lower political instability, and property security. Improved institutions boost productivity, attracting FDI [20].

Pakistan is a country that has witnessed political unrest in different forms, widespread corruption, deteriorating law and order, and other issues that appear to be barriers to attracting FDI. As foreign direct investment is regarded as a more effective factor of capital inflows to the domestic market, it is a preferred kind of capital entry to develop and less developed economies such as Pakistan. The data also show that the combined influence of institutions and trade openness has a long-term and a short-term impact on attracting additional FDI inflows [21]. The quality of the host country's institutions, macroeconomic indices, and natural resources all influence the inflow of FDI into emerging economies. Economic activities are generally better in economies with good organizations, lower governmental risk, and much more asset ownership laws. The data set, which spans the years 1984 to 2013, was applied to analyze this link. Other than institutional quality, three macroeconomic variables have been used to establish a relationship: exchange rate, per capita GDP, and natural resources using the unit root test, cointegration test, autoregressive distributed lag model, and error correction model. The result of the long-term cointegration of the variables was demonstrated using the cointegration test. Institutional quality has a considerable progressive connotation with FDI in the short and long run according to the predicted elasticities. In both the short and long term, per capita GDP has a considerable positive association with FDI. These findings recommended that an improvement in institutional quality and per capita GDP will lead to an increase in FDI inflow [22].

To determine the implications of institutional quality in FDI inflow, we reviewed several earlier research articles. The importance of FDI in boosting an economy's economic growth was addressed. As a result, every country is competing to obtain the largest proportion of global FDI. Studies have been conducted in the past to examine the impact of many factors on FDI, and one of the main factors affecting FDI influx is INQ. Different researchers have applied various criteria, such as the business climate, political stability, corruption, level of business certainty, law and order, and many other elements, in the development of the INQ index. The amount of FDI entering a country will decrease if the INQ does poorly there. INQ reforms directly improve the business climate, luring foreign

capital into the domestic market. Most studies looked at how INQ affected FDI inflow over time. The current study examined the effects of INQ in each sector, including the primary, secondary, and tertiary sectors, to determine how it affects FDI at the sectoral level in Pakistan. The policy-maker will be assisted by this information in altering the pattern of foreign investment entry into the economy to promote economic growth.

The current study evaluated the four following hypotheses regarding the fundamental relationship of the study's included variables based on the body of literature.

**Hypothesis 1.** *Institutional quality is expected to have a positive effect on foreign direct investment (FDI) in the primary sector.*

**Hypothesis 2.** *Institutional quality is expected to have a positive effect on foreign direct investment (FDI) in the secondary sector.*

**Hypothesis 3.** *Institutional quality is significantly and positively correlated with foreign direct investment (FDI) in the tertiary sector.*

### 3. Methodology

The study used time series data for Pakistan to attain a specified goal. This study was limited to a single-country analysis. A 34-year sample of data was gathered between 1986 and 2019. Two sources were used to gather the information: the World Bank and the International Country Risk Guide (Table 1) [23,24].

**Table 1.** Descriptions of variables.

| Variables | Description | Source |
|---|---|---|
| FDI Primary | Foreign Direct Investment in Agriculture and raw materials (Millions of US Dollars) | [23] |
| FDI Secondary | Foreign Direct Investment in Manufacturing (Millions of US Dollars) | [23] |
| FDI Tertiary | Foreign Direct Investment in Services (Millions of US Dollars) | [23] |
| INQ | Institutional Quality (will be developed using the following components corruption, law and order, government stability, democratic accountability, bureaucratic quality, and investment profile through PCA) | [23] |
| TO | Trade Openness (Exports plus imports divide by GDP) | [24] |
| HDI | Human Development Index | [23] |
| GDP | Gross Domestic Product (Billions of US Dollars) | [24] |

Unit root tests were applied in this study to examine each data series' unit root issue. To investigate the presence of a unit root and confirm the sequence of integration of each data series used in the study, the augmented Dickey–Fuller [25] and Phillips–Perron [26] tests were applied. Jordan and Philips introduced a more sophisticated ARDL model called the dynamic autoregressive distributed lag simulation model. This model's objective was to address the estimation problems associated with the short- and long-term model specifications of the straightforward ARDL model. This innovative model may estimate, stimulate, and robotically anticipate counterfactual changes in one explanatory variable and their consequences on explained variables while keeping other control variables constant [11,27]. This model shows how the variables are related over the short and long term by automatically estimating, producing, and plotting graphs of expected positive and negative variation in the variables. Only the relationship between short-run and long-run variables may be estimated using the ARDL model created by [27]. If the novel dynamic simulated ARDL model is appropriate, the study's variables will all be level or first difference integrated, I(0) or I(1), and stationary.

$$
\begin{aligned}
\Delta(FDI_i)_t = \alpha_o &+ \phi_o(FDI_i)_{t-1} + \mathcal{L}_1\Delta(INQ)_t + \phi_1(INQ)_{t-1} + \mathcal{L}_2\Delta(TO)_t \\
&+ \phi_2(TO)_{t-1} + \mathcal{L}_3\Delta(HDI)_t + \phi_3(HDI)_{t-1} + \mathcal{L}_4\Delta(GDP)_t \\
&+ \phi_4(GDP)_{t-1} + e_t
\end{aligned}
\tag{1}
$$

The terms foreign direct investment (FDI), institutional quality (INQ), trade openness (TO), human capital index (HDI), and gross domestic product (GDP) are used in the equation above. This equation illustrates the relationship between dependent and independent variables.

$$\Delta(FDIP)_t = \alpha_0 + \beta_0 FDIP_{t-1} + \delta_0 \Delta(INQ)_t + \beta_1(INQ)_{t-1} + \delta_1 \Delta(TO)_t \\ + \beta_2(TO)_{t-1} + \Delta\delta_2(HDI)_t + \beta_3(HDI)_{t-1} + \Delta\delta_3(GDP)_t \\ + \beta_4(GDP)_{t-1} + e_t \tag{2}$$

$$\Delta(FDIS)_t = \alpha_0 + \beta_0 FDIS_{t-1} + \delta_0 \Delta(INQ)_t + \beta_1(INQ)_{t-1} + \delta_1 \Delta(TO)_t \\ + \beta_2(TO)_{t-1} + \Delta\delta_2(HDI)_t + \beta_3(HDI)_{t-1} + \Delta\delta_3(GDP)_t \\ + \beta_4(GDP)_{t-1} + e_t \tag{3}$$

$$\Delta(FDIT)_t = \alpha_0 + \beta_0 FDIT_{t-1} + \delta_0 \Delta(INQ)_t + \beta_1(INQ)_{t-1} + \delta_1 \Delta(TO)_t \\ + \beta_2(TO)_{t-1} + \Delta\delta_2(HDI)_t + \beta_3(HDI)_{t-1} + \Delta\delta_3(GDP)_t \\ + \beta_4(GDP)_{t-1} + e_t \tag{4}$$

Foreign direct investment is the dependent variable in the abovementioned equations, whereas institutional quality is the independent variable. Trade openness, the human development index, and gross domestic product are denoted as TO, HDI, and GDP, respectively. The intercept is shown by $\alpha_0$, and the parameters are denoted by $\beta$ and $\delta$, while $e$ stands for the error term, $t$ denotes the present moment, and $t-1$ denotes the past moment or lag value. In the end, the study used a variety of diagnostic tests to examine the nonlinear ARDL model for autocorrelation, heteroskedasticity, residual normality issues, and functional form. These tests include the Ramsey Reset test for the functional form of the model, the Breusch–Godfrey serial correlation LM test, the Breusch–Pagan–Godfrey heteroskedasticity test, the Jarque–Bera test for normality, and the Breusch–Godfrey heteroskedasticity test [28]. CUSUM was employed to confirm the stability of the parameters.

## 4. Results and Discussion

The descriptive statistics for a few selected variables are shown in Table 2. This information tells us about the basic characteristics of the series. It aids in understanding the fundamental characteristics of the variables. Table 2 includes a list of the variables used in this study, including the mean, median, maximum value, minimum value, and standard deviation. The term "mean value" refers to the data's average value. The median marks the divide between the lower and upper halves of the series. Data must be sorted in either ascending or descending order because they are typically in the middle of the data. The results of this analysis suggest that there is no outlier in the data because there is no discernible difference between the variable's mean and median values. An outlier between the values of the data and the mean value is a statistic that displays significant and uneven distance. The standard deviation shows how far the data deviate from the mean (Table 2).

**Table 2.** Descriptive statistics.

| Variables | Mean | Max | Min | St. Dev |
|---|---|---|---|---|
| FDIP | 93.46 | 564.10 | 3.60 | 134.76 |
| FDIS | 593.78 | 1483.40 | 119.60 | 323.73 |
| FDIT | 130.80 | 475.50 | 3.10 | 121.23 |
| INQ | 22.10 | 31.25 | 13.08 | 3.76 |
| TO | 0.35 | 0.43 | 0.26 | 0.04 |
| HDI | 1.61 | 1.80 | 1.33 | 0.18 |
| GDP | 130 | 320.91 | 31.90 | 91.30 |

When the data's mean, variance, and covariance are not constant, the unit root problem occurs. We apply the cointegration method after first setting our data stationary. To recognize and address the unit root problem in the data, the study used two different

types of unit root tests. If we divide the data into multiple sections and compute the mean, variance, and covariance of each section, the results of each component will be different from the other portions in the case of a unit root problem. A trend (time impact) in the data is a frequent cause of unit root. The results of spurious regression, which includes variables with a unit root problem, are inaccurate and unpredictable [29]. To address the issue of stationarity, we used the augmented Dickey–Fuller [25] and Phillips–Peron (P&P) tests [26]. These tests were run on a single kind of equation: a constant and trend equation (with intercept term as well as trend variable). Table 3 gives findings of ADF and PP tests.

**Table 3.** Results of ADF and PP tests (trend and intercept).

| Variable | ADF Test | | PP Test | |
| --- | --- | --- | --- | --- |
| | I(0) | I(1) | I(0) | I(1) |
| LnFDIP | 4.79 | −5.36 *** | −3.90 ** | - |
| LnFDIS | −2.62 | −6.97 *** | −2.96 | −6.97 *** |
| LnFDIT | −3.57 ** | - | −4.34 *** | - |
| lnINQ | −4.69 *** | - | −5.11 *** | - |
| LnTO | −2.39 | −6.29 *** | −1.22 | −7.00 *** |
| LnHDI | −1.81 | −3.92 *** | −1.91 | −4.12 *** |
| LnGDP | −1.96 | −5.29 *** | −2.01 | −5.22 *** |

Note: *** and ** indicate 1% and 5% levels of significance, respectively.

The variables FDIP primary, FDIS secondary, HDI, GDP, and TO are nonstationary at level I in the results of the ADF test in this study, while, FDIT tertiary, and INQ are stationary at level I(0), meaning without difference (0).

Three variables—FDIP primary, FDIT tertiary, and INQ—are stationary at level I(0) in the results of the PP test in this study, but the other selected variables—FDIS secondary, HDI, GDP, and trade openness—are nonstationary at level I(0) (zero difference). Therefore, following an integrated series process, all variables are stationary at the first difference, I(1) (Table 3).

This study's F-statistic value is highly significant at 1% since it exceeds the upper-bound value in every case (4.41 > 4.44), (9.31 > 4.44), (8.42 > 4.44), and (7.92 > 4.44). This demonstrates that we adopt alternative hypothesis H 1, which demonstrates the presence of cointegration in the chosen dynamic simulated ARDL model. This implies that INQ has a long-term effect on sectoral FDI in Pakistan. The results of the F-bound test employed in this study are shown below (Table 4).

**Table 4.** Findings of the F-bounds cointegration test.

| LnFDI Primary Sector | | LnFDI Secondary Sector | | LnFDI Tertiary Sector | |
| --- | --- | --- | --- | --- | --- |
| Test Statistic | F value | Test Statistic | F Value | Test Statistic | F Value |
| F-stat | 9.31 *** | F-stat | 8.42 *** | F-stat | 7.92 *** |
| Critical bounds value Significance | Lower Bound L0 | Upper Bound L1 | | | |
| 10% | 2.12 | 3.23 | | | |
| 5% | 2.45 | 3.61 | | | |
| 1% | 3.16 | 4.44 | | | |

Note: *** indicate 1% significance.

Table 5 displays the outcomes of the dynamically stimulated ARDL model for Pakistan. The results indicate a sector-by-sector link between INQ and FDA. The outcome suggests that INQ has a significant and advantageous impact on sectoral FDI in Pakistan [20,22]. It demonstrates two sorts of relationships between the variables: a long-term relationship and a short-term relationship. We start by outlining the long-term connection. Regarding

FDI, the primary sector INQ has a positive and important influence at 5%, while other model variables (TO $_t$ and TO $_{t-1}$ are at 5%. At 5%, the impact of its lag value on FDI is significant. The INQ effect in the FDI secondary sector is significant at 10%, and the model's other variables (TO$_t$, TO$_{t-1}$) are positively significant at 5%. The substantial effects of HDI and FDIS$_{t-1}$ are significant at 10% and 5%, respectively. In the most recent instance, the INQ has a 5% positive relationship with tertiary FDI. At 5% and 10%, variables such as GDP, HDI, respectively. The values of TO$_t$ and TO$_{t-1}$ are inconsequential (Table 5).

**Table 5.** Results of dynamic simulated ARDL.

| Variables | FDI Primary | FDI Secondary Sector | FDI Tertiary Sector |
|---|---|---|---|
| **Long Run** | | | |
| $Ln\ INQ_t$ | 0.153 ** | 0.148 * | 0.124 ** |
| | (0.036) | (0.072) | (0.037) |
| $Ln\ TO_t$ | 0.105 ** | 0.075 ** | 0.171 |
| | (0.049) | (0.036) | (0.251) |
| $Ln\ TO_{t-1}$ | 0.068 ** | 0.089 ** | 0.190 |
| | (0.031) | (0.044) | (0.341) |
| $Ln\ HDI_t$ | 0.147 | 0.135 * | 0.140 * |
| | (0.152) | (0.057) | (0.062) |
| $Ln\ GDP_t$ | 0.617 | 0.117 * | 0.118 ** |
| | (0.241) | (0.058) | (0.038) |
| $Ln\ FDIP_{t-1}$ | 0.096 ** | | |
| | (0.025) | | |
| $Ln\ FDIS_{t-1}$ | | 0.103 ** | |
| | | (0.025) | |
| $ln\ FDIT_{t-1}$ | | | 0.048 ** |
| | | | (0.017) |
| **Short run** | | | |
| $Ln\ \Delta INQ_t$ | 0.146 * | 0.156 * | 0.089 ** |
| | (0.056) | (0.063) | (0.019) |
| $Ln\ \Delta TO_t$ | 0.133 * | 0.113 ** | 0.105 * |
| | (0.054) | (0.042) | (0.051) |
| $Ln\ \Delta TO_{t-1}$ | 0.092 ** | 0.086 ** | 0.114 ** |
| | (0.037) | (0.033) | (0.045) |
| $Ln\ \Delta HDI_t$ | 0.147 ** | 0.152 * | 0.121 ** |
| | (0.043) | (0.074) | (0.032) |
| $Ln\ \Delta GDP_t$ | 0.079 * | 0.069 ** | 0.095 ** |
| | (0.031) | (0.028) | (0.046) |
| $Ln\ \Delta FDIP_{t-1}$ | 0.124 ** | | |
| | (0.048) | | |
| $Ln\ \Delta FDIS_{t-1}$ | | 0.191 ** | |
| | | (0.048) | |
| $Ln\ \Delta FDIT_{t-1}$ | | | 0.184 * |
| | | | (0.057) |
| Constant | 0.176 ** | 0.193 * | 0.165 * |
| | (0.039) | (0.095) | (0.079) |

Note: ** and * indicate 5% and 10% significance, respectively.

The F-bound test is applied to examine the overall model's cointegration. In this study, the value of the F-statistic in all cases is greater than the upper bound value at 1%, i.e., 4.41 > 4.44, 9.31 > 4.44, 8.417 > 4.44, and 7.915 > 4.44, which shows that the F-statistic is highly significant at 1%. This shows that we accept the alternative hypothesis $H_1$, showing the existence of cointegration in the selected dynamic simulated ARDL model. In the case of Pakistan, this means that the INQ has a long-term impact on sectoral FDI. The decision might also be made by comparing the F-statistical/F-calculated value with the F-tabulated/F-critical value provided in [10]. If the F-statistical value exceeds the F-tabulated value in the Pesaran table, we accept the alternative hypothesis $H_1$, indicating that cointegration exists in the selected dynamic simulated ARDL model or vice versa.

Reference [30] suggested a simulated ARDL approach to investigate the short- and long-term association between dependent and independent variables. It can simulate, estimate, and plot counterfactual change prediction in a regression in the dependent variable while keeping the other variables constant.

The results of the dynamically stimulated ARDL model for Pakistan are shown in Table 5. The results show the relationship between INQ and FDI by sector. The result indicates that INQ has positive and significant effect on sectoral FDI in Pakistan [20,21]. It shows two types of relationships between the variables: one is a long-run relationship, and the second is a short-run relationship. First, we explain the relationship in the long run. In the case of FDI, primary sector INQ has a positive and significant role at 5%, and other variables included in the model ($TO_t$ and $TO_{t-1}$) have a 5% and positively significant effect on FDI. The INQ effect is significant in the FDI secondary sector at 10%, and the other variables of the model ($TO_t$, $TO_{t-1}$) are positively significant at 5%. HDI and GDP have significant effects at 10%, respectively. In the last case, the INQ has a positive relation with FDI tertiary at 5%. Variables (GDP, HDI) have a significant effect at 5% and 10%, respectively. However, the $TO_t$ and $TO_{t-1}$ values are insignificant. In the second type of relation, we analyzed the short-run relationship between the variables. In the short run, INQ and other included variables (TO, $TO_{t-1}$, GDP, HDI) have a significant effect on FDI in the primary, secondary and tertiary sectors.

Market connectedness is harmed by insufficient institutions, which obstruct trade flows, cause market frictions, and cause unnecessary delays. As a result, the actual cost of manufacturing rises, reducing FDI inflow into the domestic economy. Weak institutional quality reduces an economy's competitive advantage, but the accessibility of high-quality institutions increases its comparative advantage, both internationally and domestically. Gross domestic product, human development index, and trade openness alongside the main variables have favorable and significant implications for institutional quality and FDI inflow, both collectively and separately, implying that for improvement in FDI inflow in Pakistan, a revised open policy for more development of the institutional quality process is necessary. Maintaining an open and transparent policy, as well as improving the quality of Pakistan's institutional system, is also critical for the country's FDI attraction [30].

The latest empirical findings support the notion that foreign investors are drawn to that institutional quality because it lowers application expenses and makes doing business in the host country simpler. Poor institutions, on the other hand, obstruct FDI and might operate as a tax, raising the cost of FDI [31]. Since these factors raise the cost of doing business [32,33], investors are wary of making investments in countries where institutions encourage corruption, nepotism, and bureaucracy. Weak institutions impair market connection, obstructing the realization of trade potential, causing market frictions and unnecessary delays, and thus raising the production cost, which has a negative impact on FDI influx into the home economy. Providing a structure of efficient quality institutions boosts an economy's competitive advantage, both on the national and international fronts, whereas the availability of poor-quality institutions reduces it. Economies with competent institutions are more likely to fascinate FDI, whereas those with less competent institutions are less likely to attract FDI flow and financial interdependence.

The reliability, stability, and predictability of the data were assessed using a range of diagnostic procedures. Table 6 displays the findings of the diagnostic examinations. To find the heteroscedasticity issue in the model, the Breusch–Pagan LM test is used. That residual variance is not constant is demonstrated by the heteroscedasticity problem. The chi-squared (2) distribution serves as its foundation. We accept the alternative hypothesis $H_1$, which states that the model has an issue with heteroscedasticity if the probability value of the 2-statistic is less than 5% (0.05), and we accept the null hypothesis $H_0$, which states that the data have homoscedasticity if the probability value of the 2-statistic is greater than 5% (>0.05), showing that there is no problem of heteroscedasticity. According to our results in the case of all three models, FDIP, FDIS and FDIT, the statistical values are greater than 5% (0.119, 0.276, 0.248 > 0.05). Therefore, we accept the null hypothesis $H_0$, indicating that heteroscedasticity does not exist. A test known as the Breusch–Godfrey LM [34] is used to pinpoint the model's serial correlation issue. Additionally, serial correlation shows that there is a correlation between the residuals of two time periods, though in this instance, only between the residual terms of two subsequent time periods. The decision-making procedure is as follows: if the probability of the calculated value is less than 5% (0.05), we accept the alternative hypothesis $H_1$, which indicates that serial correlation is a problem in the model; if the probability of the calculated value is greater than 5% (>0.05), we accept the null hypothesis $H_0$, which indicates that serial correlation is not a problem in the data. The probability values in our models (FDIP, FDIS, and FDIT) are 0.747, 0.060, and 0.061, respectively, which are all more than 5% (0.747, 0.060, and 0.061 > 0.05). This demonstrates that we accept the null hypothesis $H_0$, which says that the serial correlation issue is not present in the dynamic simulated ARDL model that has been implemented (Tables 6 and 7).

**Table 6.** Results of the diagnostic test (FDI primary).

| Econometric Issues | Test Names | F-Statistics | *p* Values |
| --- | --- | --- | --- |
| Heteroscedasticity | Breusch–Pagan LM [28] | 2.01 | 0.12 |
| Serial Correlation | Breusch–Godfrey LM [34] | 0.29 | 0.75 |
| Normality | Jarque–Bera [35] | 3.06 | 0.15 |
| Specification | Ramsey RESET [36] | 1.75 | 0.20 |

**Table 7.** Results of the diagnostic test (FDI secondary).

| Econometric Issues | Test Names | F-Statistics | *p* Values |
| --- | --- | --- | --- |
| Heteroscedasticity | Breusch–Pagan LM [28] | 1.35 | 0.28 |
| Serial Correlation | Breusch–Godfrey LM [34] | 2.59 | 0.16 |
| Normality | Jarque–Bera [35] | 2.02 | 0.14 |
| Specification | Ramsey RESET [36] | 2.45 | 0.12 |

The Jarque–Bera test [35] was additionally employed to guarantee the data's normal distribution. The data are equally distributed on both sides of the mean value, according to the normal distribution. The F-distribution is followed by the Jarque–Bera test. The decision rule is that we accept the alternative hypothesis $H_1$, indicating that the data are not normally distributed if the probability value of the F-statistic is less than 5% (0.05), and we accept the null hypothesis $H_0$, indicating that the data are normally distributed if the probability value of the F-statistic is greater than 5% (>0.05). The probability values of the F-statistic in our models (FDIP, FDIS, and FDIT) in the results of this study are 0.149, 0.140, and 0.1563, respectively, which are larger than 5% (i.e., 0.149, 0.140, and 0.1563 > 0.05). It demonstrates that we accept the null hypothesis $H_0$, which claims that the distribution of our data is normal. Additionally, Ramsey RESET (regression equation specification error test) was used in the study to assess the accuracy of the model specification [36]. It determines whether the model's chosen lag length is the most suitable. It is F-distribution-compliant. The decision rule is that we accept the alternative hypothesis $H_1$, indicating that the chosen model is not the most appropriate model if the probability value of the F-statistic

is less than 5% (0.05), and we accept the null hypothesis $H_0$, indicating that the model used in the study is the most appropriate if the probability value of the F-statistic is greater than 5% (>0.05). In the outcomes of this research, the probability values of the F-statistic are 0.196, 0.116 and 0.1790, respectively, which are greater than 5% (i.e., 0.196, 0.116 and 0.1790 > 0.05). It shows that we will accept the null hypothesis $H_0$, which confirms that the researcher has applied the most appropriate ARDL model in this study (Table 8).

**Table 8.** Results of the diagnostic test (FDI tertiary).

| Econometric Issues | Test Names | F-Statistics | $p$ Values |
|---|---|---|---|
| Heteroscedasticity | Breusch–Pagan LM [28] | 1.44 | 0.25 |
| Serial Correlation | Breusch–Godfrey LM [34] | 2.49 | 0.16 |
| Normality | Jarque–Bera [35] | 3.71 | 0.16 |
| Specification | Ramsey RESET [36] | 2.23 | 0.18 |

The results of the CUSUM test are shown in Figures 1–3. This test was used to see if the model's parameters were stable [35]. In these tests, the significance threshold is calculated at 5%. The alternative hypothesis $H_1$ demonstrates that the parameters utilized in this research are stable is accepted if the results of both methodologies are significant at a 5% level. If the results of both tests are not statistically significant at 5%, it is accepted that the null hypothesis $H_0$ demonstrating the instability of the chosen parameters is true. In the case of FDI, FDI primary, FDI secondary and FDI tertiary are explained in Figures 1–3, respectively, which demonstrates that the results of the CUSUM test are significant at 5%. Since the parameters utilized in the ARDL F-bound test technique are stable, we accept the alternative hypothesis $H_1$ (Figures 1–3).

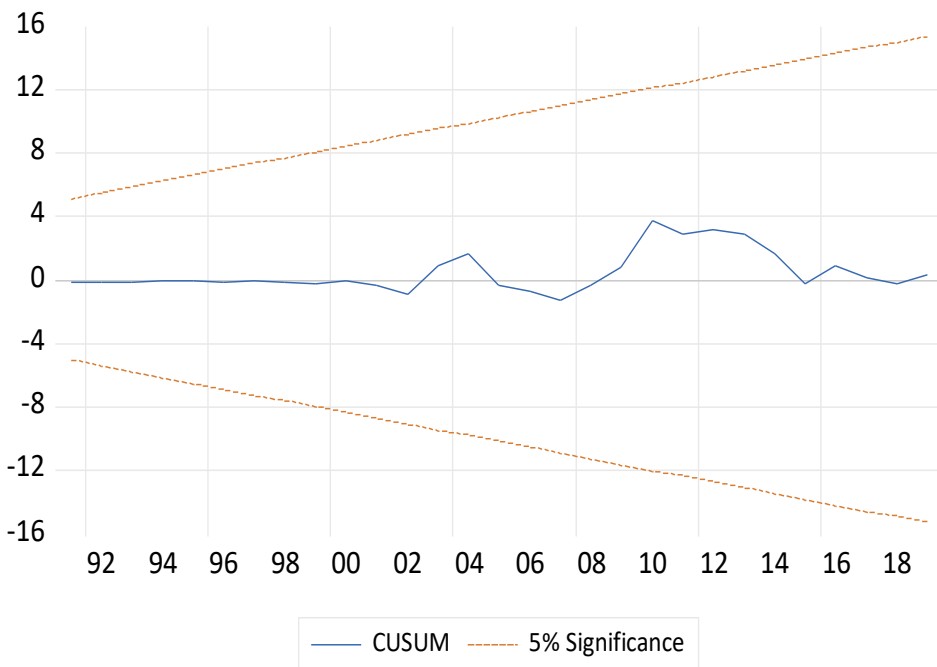

**Figure 1.** Findings of CUSUM test.

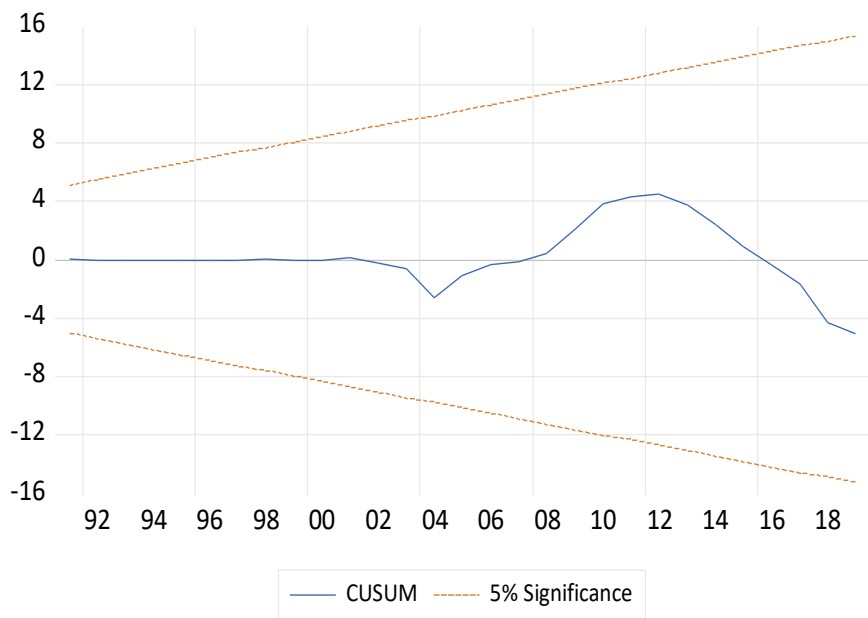

**Figure 2.** Findings of CUSUM test.

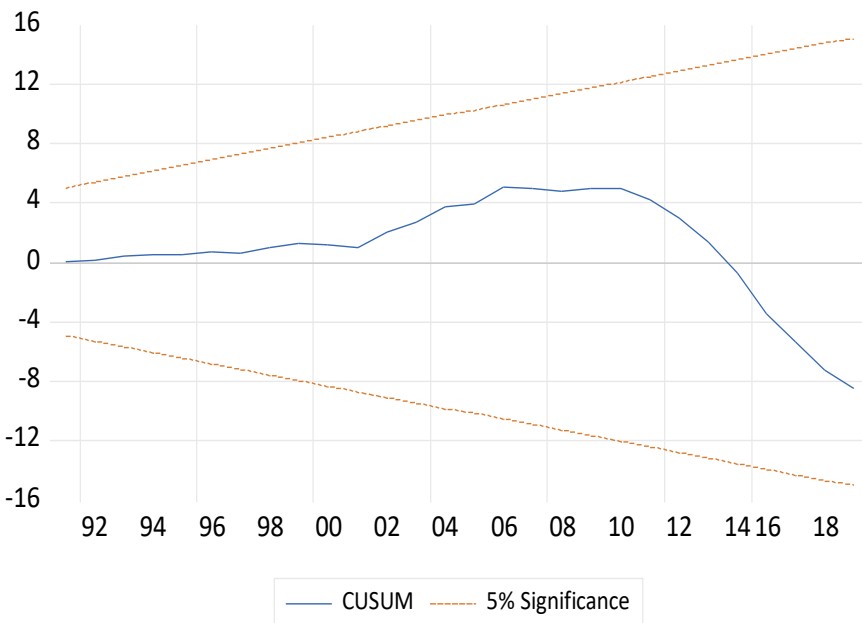

**Figure 3.** Findings of CUSUM test.

## 5. Conclusions and Policy Implications

The main goal of this study was to examine the connection between Pakistani sectoral FDI and institutional quality. This study is limited to a single-country analysis. The sample size of the data was selected as 1986 to 2019. The results of the dynamically stimulated ARDL model show the relationship between INQ and FDA sectors. In the case of FDI, primary sector INQ has a positive and significant role in the long run. In the short run, INQ and other included variables (TO, $TO_{t-1}$, GDP, HDI) have a significant effect on FDI in the primary, secondary and tertiary sectors. In [37], many significant linkages of institutional quality and FDI inflows, as well as flow of FDI and development due to productivity increase, were discovered. The exact extent of the linkages between emerging and advanced economies is still unknown. The institutional quality and FDI–efficiency connection is highlighted inside a combined theoretical general equilibrium outline in this

research. It creates an explicit separation of the two economies, although local and overseas businesses utilize various tools to compete on the market for finished goods.

The findings of the study conclude that higher institutional quality in emerging governments such as Pakistan encourages major technological advancements transfers through FDI, increasing the overall performance of the economy. Firms have substantial new incentives to engage in FDI because of recent institutional advances in many emerging countries, as well as the incredible growth of transportation and communication technology over the last few decades. They created an FDI model that included heterogeneous enterprises and people using various knowledge sources, and they investigated the relationship between INQ and FDI productivity using a unified theoretical general equilibrium method and institutional quality-adapted FDI costs. Reference [38] explains that deficiencies in institutional infrastructure were found during the 1997 financial crisis in Asia, and institutional reforms took place in many countries to overcome this weakness to encourage more FDI inflow and to maximize its positive effects on the economy. World FDI moves towards those countries that are practicing high standards of institutional quality, while countries with weak institutional performance face difficulties attracting FDI. The influence of institutions on FDI may be analyzed in a few ways. The weak performance of institutions discourages FDI inflow, such as a tax on imports. Inefficient institutions generate uncertainty in the market, which makes profit doubtful for foreign investors and indirectly deters capital inflow.

The research showed that INQ plays a key role in increasing FDI inflow in the country. Moreover, our model examined the impact of INQ on sectors of FDI (primary, secondary, and tertiary). The results showed that INQ affects all the subsectors of FDI in Pakistan. This means that as the INQ improves, it attracts more FDI inflow in Pakistan in all sectors. Maintaining an open and progressive policy, as well as improving the quality of Pakistan's institutional system, is also essential for attracting FDI. To maximize the impact of the policy combination, policies for institutional change and informal activity elimination should be implemented together. To stimulate economic growth and increase income per capita, concurrent measures to encourage FDI through institutional quality improvement should focus on important factors, such as democratic accountability.

## 6. Limitations of Study

To encompass all the crucial aspects of institutional quality, the developed index considered six factors: corruption, law and order, government stability, democratic accountability, bureaucratic quality, and investment profile. Altering the factors or increasing their number could lead to broad empirical results. The single-country analysis is the limitation of this study. The relation between INQ and FDI can also be examined by applying the panel data technique or by increasing the size of the data set.

**Author Contributions:** Conceptualization, Z.U.K., A.K. and D.K; methodology, Z.U.K. and A.K.; software, Z.U.K., A.K. and D.K.; validation, Z.U.K., A.K., D.K. and R.M.; formal analysis, Z.U.K. and A.K.; investigation, A.K. and D.K.; resources, R.M.; data curation, Z.U.K., A.K. and D.K.; writing—original draft preparation, Z.U.K. and A.K.; writing—review and editing, A.K., D.K. and R.M.; visualization, A.K. and D.K.; supervision, A.K. and D.K.; project administration, A.K. and D.K.; funding acquisition, R.M. All authors have read and agreed to the published version of the manuscript.

**Funding:** This research received no external funding.

**Institutional Review Board Statement:** Not applicable.

**Informed Consent Statement:** Not necessary.

**Data Availability Statement:** Data are openly accessed and freely available to everyone.

**Acknowledgments:** This research is a part of a research thesis submitted to Kohat University of Science & Technology (KUST), Kohat Khyber Pakhtunkhwa.

**Conflicts of Interest:** The authors declare no conflict of interest.

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
