# Peer review of "The Impact of Institutional Quality on Sectoral Foreign Direct Investment in Pakistan: A Dynamic Simulated ARDL Approach"

_sustainability, doi:10.3390/su15097231_

Round 1

Reviewer 1 Report

The issues raised in the article are up-to-date and constitute an important problem both in economic theory and in business practice. The structure of the article is well structured.

In the Introduction, the authors in a synthetic way justify why they use the ARDL method. They also characterize this method in a general way. The statistical tools used in the empirical part were properly selected. The use of appropriate methods allowed us to obtain interesting results. The authors refer to the current literature. They cite similar studies conducted so far.

The question to consider is whether it would be worth adding a paragraph in the "Introduction" on the subject of a brief description of the most important macroeconomic parameters of Pakistan in order to familiarize the recipient with the economic situation of this country. In addition, it may be worth adding a justification why Pakistan was selected for the analysis and not another country.

Author Response

Comments and response  reviewer   1

Sr. No

Reviewer Comments

Authors Response

1

The question to consider is whether it would be worth adding a paragraph in the "Introduction" on the subject of a brief description of the most important macroeconomic parameters of Pakistan in order to familiarize the recipient with the economic situation of this country. In addition, it may be worth adding a justification why Pakistan was selected for the analysis and not another country

Thank you for your suggestions. This comment has been addressed in the introduction section of the maunscript

Reviewer 2 Report

Article title:  The Impact of Institutional Quality on Sectoral Foreign Direct Investment in Pakistan: A Dynamic Simulated ARDL Approach

After reviewing this paper carefully, I have some comments below:

- In the introduction, the authors need to clearly show the research gap, what does this study contribute to the literature? In this section, I see that the authors have made a good effort in presenting the motivation but need to add the above issues.

- In the literature review, the reviewers are very sketchy. Actually, I only see the author review and cite 3 previous research. The authors need to supplement studies related to FDI and the role of institutional quality. Therefore, I recommend authors review and cite studies such as Dang et al (2021); Almustafa et al. (2023); Nguyen (2022); Islam et al (2020) Dang et al. (2022); Nguyen et al (2022)… (see reference)

- Authors need to get the number of decimals uniform across all tables. I see some numbers take 2, some take 1 decimal.

- Section 5 should be Conclusions, Policy Implications, and Limitations. The authors need to add research limitations and directions for further research.

- There are some grammatical errors, which the authors need to check carefully

In general, this study can be published if the authors edit according to the above comments. In particular, the literature of this article is very weak, the authors need to carefully review and edit it as I suggest.

 References

Almustafa, H. et al. (2023). The impact of COVID-19 on firm risk and performance in MENA countries: Does national governance quality matter? PloS one, 18(2), e0281148.

Dang, V. C., & Nguyen, Q. K. (2021). Determinants of FDI attractiveness: Evidence from ASEAN-7 countries. Cogent Social Sciences, 7(1), 2004676.

Dang, V. C. et al. (2022). Corruption, institutional quality and shadow economy in Asian countries. Applied Economics Letters, 1-6.

Nguyen, Q. K. (2022). Audit committee structure, institutional quality, and bank stability: evidence from ASEAN countries. Finance Research Letters, 46, 102369.

Islam, M. A., Khan, M. A., Popp, J., Sroka, W., & Oláh, J. (2020). Financial development and foreign direct investment—The moderating role of quality institutions. Sustainability, 12(9), 3556.

Nguyen, Q. K., et al (2022). Does the country’s institutional quality enhance the role of risk governance in preventing bank risk? Applied Economics Letters, 1-4.

Author Response

Comments and response Second reviewer

Sr. No

Reviewer Comments

Authors Response

1

In the introduction, the authors need to clearly show the research gap.

Thank you for your suggestions. The research gap has been added to the article.

2

Reviewer Recommended some studies to review it. 

Almustafa, H. et al. (2023). The impact of COVID-19 on firm risk and performance in MENA countries: Does national governance quality matter? PloS one, 18(2), e0281148.

Dang, V. C., & Nguyen, Q. K. (2021). Determinants of FDI attractiveness: Evidence from ASEAN-7 countries. Cogent Social Sciences, 7(1), 2004676.

Dang, V. C. et al. (2022). Corruption, institutional quality and shadow economy in Asian countries. Applied Economics Letters, 1-6.

Nguyen, Q. K. (2022). Audit committee structure, institutional quality, and bank stability: evidence from ASEAN countries. Finance Research Letters, 46, 102369.

Nguyen, Q. K., et al (2022). Does the country’s institutional quality enhance the role of risk governance in preventing bank risk? Applied Economics Letters, 1-4.

Recommended literature review papers have been added to the literature

3

Authors need to get the number of decimals uniform across all tables.

All the figures are checked to make uniform according to decimals.

4

The authors need to add research limitations and directions for further research.

Limitations of the study is now included in the study.

5

There are some grammatical errors, which the authors need to check carefully.

Grammatical errors have been checked and rectified

Reviewer 3 Report

please find my comments in the attached file

Author Response

Comments and response Third Reviewer

Sr. No

Reviewer Comments

Authors Response

1

Repeating of similar text again and again in the abstract.

Add the novelty of study in abstract.

JEL classification is missing.

Thank you for your suggestions. Repeated texts are replaced by accurate words in the abstract.

Novelty is now mentioned in abstract.

JEL classification has been added

2

. The role foreign direct investment plays in helping developing nations to develop is 32 highlighted in recent research” I do not get your point that actually you want to say. Really confusing line. Please do more focus on enhancing the quality of communication

 This statement has been rephrased and the author have tried to enhance the quality of communication

3

Many statements require the literature reference. Please adequately cite the relevant literature Over time, human priorities change, requiring 35 the introduction of new technologies, made possible by FDI, who said this?

Literature support has been added

4

Please add one table of literature review summary relating to topic of paper specifically in the case of Pakistan.

Added the literature review related to Pakistan in paragraph form to the literature section

5

African nations are working to solve y was conducted on Pakistan. Please offer some statistics in the case of Pakistan that motivate for further reading.

This statement has been omitted from the introduction part

6

The introduction part is too dry and unable to convince the reader for reading. Please clearly focus on the problem statement, rationale of study, significance, and background of study.

Introduction part has been rearranged as per the reviewer comments

7

Please add some empirical studies in literature review supporting the relationship between institutional quality and FDI, I dot find any hypothesis in this section

Empirical studies related to the topic have been added.

Hypotheses have been added

8

. Please clearly elaborate the gap in existing literature and how this study filled this gap.

Research gap has been added

9

Please support your discussion of results with recent studies. Are results align with existing studies? if yes, then what is the contribution, if not, why it departs. Please clearly state both.

Literature support have been added to the results discussion

10

Please add the limitation of study

Limitations of study have been added.

Round 2

Reviewer 2 Report

This version is better and can be published

Author Response

Thanks for your valuable comments. There is no such comments that need to be incorporated in the revised manuscript. 

Reviewer 3 Report

the author can get some guidance from this article.

Farooq, . Foreign direct investment, foreign aid, and CO2 emissions in Asian economies: does governance matter?. Environ Sci Pollut Res 29, 7532–7547 (2022). https://doi.org/10.1007/s11356-021-16115-3

Author Response

The reviewer has mentioned one paper for the guidance. we have read that paper and got the guidance from that. other than that, there is no such comments that need to incorporate in the revised draft.